# Polyphenols from *Maackia amurensis* Heartwood Protect Neuronal Cells from Oxidative Stress and Prevent Herpetic Infection

**DOI:** 10.3390/ijms25084142

**Published:** 2024-04-09

**Authors:** Darya V. Tarbeeva, Evgeny A. Pislyagin, Ekaterina S. Menchinskaya, Dmitrii V. Berdyshev, Natalya V. Krylova, Olga V. Iunikhina, Anatoliy I. Kalinovskiy, Mikhail Y. Shchelkanov, Natalia P. Mishchenko, Dmitry L. Aminin, Sergey A. Fedoreyev

**Affiliations:** 1G.B. Elyakov Pacific Institute of Bioorganic Chemistry, Far-Eastern Branch of the Russian Academy of Science, 690022 Vladivostok, Russia; pislyagin_ea@piboc.dvo.ru (E.A.P.); menchinskaya_es@piboc.dvo.ru (E.S.M.); berdyshev@piboc.dvo.ru (D.V.B.); kaaniw@piboc.dvo.ru (A.I.K.); mischenkonp@mail.ru (N.P.M.); daminin@piboc.dvo.ru (D.L.A.); fedoreev-s@mail.ru (S.A.F.); 2G.P. Somov Institute of Epidemiology and Microbiology, Rospotrebnadzor, 690087 Vladivostok, Russia; krylovanatalya@gmail.com (N.V.K.); olga_iun@inbox.ru (O.V.I.); adorob@mail.ru (M.Y.S.)

**Keywords:** polyphenolic compounds, HSV-1, virucidal activity, neuroprotective activity, ROS

## Abstract

Here, we continued the investigation of anti-HSV-1 activity and neuroprotective potential of 14 polyphenolic compounds isolated from *Maackia amurensis* heartwood. We determined the absolute configurations of asymmetric centers in scirpusin A (**13**) and maackiazin (**10**) as 7*R*,8*R* and 1″*S*,2″*S*, respectively. We showed that dimeric stilbens maackin (**9**) and scirpusin A (**13**) possessed the highest anti-HSV-1 activity among polyphenols **1**–**14**. We also studied the effect of polyphenols **9** and **13** on the early stages of HSV-1 infection. Direct interaction with the virus (virucidal activity) was the main mechanism of the antiviral activity of these compounds. The neuroprotective potential of polyphenolic compounds from *M. amurensis* was studied using models of 6-hydroxydopamine (6-OHDA)-and paraquat (PQ)-induced neurotoxicity. A dimeric stilbene scirpusin A (**13**) and a flavonoid liquiritigenin (**6**) were shown to be the most active compounds among the tested polyphenols. These compounds significantly increased the viability of 6-OHDA-and PQ-treated Neuro-2a cells, elevated mitochondrial membrane potential and reduced the intracellular ROS level. We also found that scirpusin A (**13**), liquiritigenin (**6**) and retusin (**3**) considerably increased the percentage of live Neuro-2a cells and decreased the number of early apoptotic cells. Scirpusin A (**13**) was the most promising compound possessing both anti-HSV-1 activity and neuroprotective potential.

## 1. Introduction

The leguminous woody plant *Maackia amurensis* Rupr et Maxim is widespread in the far east of the Russian Federation, particularly in Primorsky Krai. *M. amurensis* heartwood is a rich source of isoflavones, pterocarpans, flavanones, isoflavans, and isoflavanones. The heartwood of this plant contains not only isoflavonoids widely distributed in the Leguminosae such as genistein, daidzein, formononetin, tectorigenin, but also monomeric and dimeric stilbens, as well as rare isoflavonostilbenes and stilbenolignans [1]. These polyphenols constitute the polyphenolic complex of *M. amurensis* heartwood. In contrast to the heartwood, the roots of this plant produce isoflavone glycosides and prenylated flavanones [2,3].

The polyphenolic complex from *M. amurensis* heartwood possessed significant hepatoprotective activity and was registered in the Russian Federation as a drug named Maksar^®^ [1]. In addition to hepatoprotective properties, Maksar^®^ also possessed antithrombogenic, antiplatelet, antitumor, and antioxidant properties [4,5,6].

The therapeutic efficacy of Maksar^®^ was shown when administrated to patients with chronic hepatitis including viral hepatitis B and C [7]. The authors suggested that the antiviral effect of Maksar^®^ was due to the activation of immune response. A recent report revealed the anti-SARS-CoV-2 activity of Maksar^®^ [8]. This effect can be due to the ability of polyphenols from Maksar^®^ maackolin, maackin, and scirpusin A to bind to the main Mpro protease of SARS-CoV-2 predicted using molecular docking approach [9,10].

In addition to anti-SARS-CoV-2 activity, we found that Maksar^®^ inhibited the replication of herpes simplex virus type 1 (HSV-1) and enterovirus B (ECHO-1) [11]. Herpesviruses can penetrate into the central nervous system and play a role in the pathogenesis of neurodegenerative disorders [12,13]. HSV-1 was reported to cause an increase in the level of intracellular reactive oxygen species (ROS) and lipid peroxidation. The virus also reduced the content of glutathione, which plays a crucial role in the body’s antioxidant defense [14,15]. Additionally, HSV-1 infection induced a remodeling of mitochondrial shape [16]. HSV-1 was shown to bind to mitochondrion-associated factors. These interactions altered mitochondrial dynamics in neurons, thereby facilitating viral replication and pathogenesis [17]. There is a clear connection between herpes infection and the subsequent development of neurodegenerative diseases, which is largely due to the virus-induced accumulation of reactive oxygen species in neurons [18]. Impaired mitochondrial function results in elevated ROS levels and subsequently leads to the development of neurodegenerative diseases [19,20,21]. The activation of inflammatory processes and host immune responses caused chronic damage resulting in alterations of neuronal function and viability such as oxidative stress, protein aggregation, deficient autophagic processes, and neuronal death [15]. Therefore, there is a need for biologically active compounds possessing both antiviral and neuroprotective properties for the treatment of herpetic infections and neurodegenerative disorders.

Maksar^®^ was shown to possess antiviral and neuroprotective properties, but it was unclear which individual compounds were responsible for these activities [11,22]. Recently, we started the study of antiherpetic and neuroprotective properties of polyphenols from *M. amurensis* in order to identify the most active compounds [22]. Here, we continue our investigation of anti-HSV-1 activity and neuroprotective potential of isoflavonoids and stilbenes from *M. amurensis*.

## 2. Results

### 2.1. Polyphenolic Compounds from M. amurensis Heartwood

We isolated 14 previously known compounds from *M. amurensis* heartwood: isoflavonoids genistein (**2**), retusin (**3**), maackiain (**4**), tectorigenin (**5**), medicarpin (**7**), formononetin (**8**), (±)-3-hydroxyvestitone (**12**), flavonoid liquiritigenin (**6**), monomeric stilbenes resveratrol (**1**), piceatannol (**11**), dimeric stilbens maackin (**9**), scirpusin A (**13**), isoflavonostilbenes maackiazin (**10**), and maackiapicevestitol (**14**) (Figure 1). We used ^1^H NMR and HPLC-MS techniques for structural identification of the isolated polyphenols.

### 2.2. Determination of Absolute Configurations for Maackiazin (***10***) and Scirpusin A (***13***)

We also determined the absolute configurations for scirpusin A (**13**) and maackiazin (**10**).

First of all, the vicinal spin–spin coupling constants J (H1″,H2″) were calculated using the B3LYP/6-311G(d)_PCM//B3LYP/6-311G(d)_PCM method for different conformations of 1″*R*,2″*S*-maackiazin (**10**) and 1″*S*,2″*S*-maackiazin (**10**). These significant conformations were preliminary selected during a conformational analysis performed at B3LYP/6-311G(d)_PCM level of theory (Appendix A). The calculated mean values of J_calc_ (H1″,H2″) were 3.8 Hz and 7.8 Hz for 1″*R*,2″*S* and 1″*S*,2″*S*-maackiazin (**10**), respectively. The experimental value of J (H1″,H2″) was 7.9 Hz, which correlated well with the calculated value for 1″*S*,2″*S*-maackiazin (**10**) (Appendix A). In order to choose between 1″*S*,2″*S*-maackiazin (**10**) and 1″*R*,2″*R*-maackiazin (**10**) options, we compared the calculated ECD spectra for these stereoisomers with the experimental one (Figure 2).

The theoretical ECD spectra calculated for different conformations of 1″*S*,2″*S*-maackiazin (**10**) contained many bands at λ ≥ 240 nm. We found that at λ ≥ 220 nm the shapes of ECD spectra were strong functions of several large-amplitude motion (LAM) processes—internal rotations of anisol and polyphenol fragments around C-C bonds and the inversion of the 1,4-dioxane cycle. Thus, the internal rotation of the anisole substituent around C(1′)-C(3) bond resulted in the pronounced changes of the ECD spectrum at λ ≥ 220 nm—many bands changed their intensity and even their sign. However, in the region 195 ≤ λ ≤ 220 nm the positions and signs of the intensive positive and intensive negative bands did not change (negative band in the region 200 ≤ λ ≤ 240 nm was a doublet). The inversion of 1,4-dioxane cycle led to the conformation, where both polyphenol substituents were in axial orientation relative to the molecular plane: θ1(H1″,H2″) ≡ ∠H1″-C1″-C2″-H2″ ≈ 81°; θ2 ≡ ∠C7-C1″-C2″-C8 ≈ −162°. For this conformation, the intensity of the high-frequency part of the negative band’s doublet in the region 195 ≤ λ ≤ 220 nm increased approximately twice, whereas the long-wave part disappeared. In the λ ≥ 250 nm region, the broad positive band changed its sign. This conformation was minor, and its total portion was approximately 1%. The rotation of polyphenol substituents resulted in variations of intensities of bands in the λ ≤ 240 nm region without changing their signs (Appendix A). Thus, the shape of the ECD spectrum of 1″*S*,2″*S*-maackiazin (**10**) was stable during these LAM processes.

The data presented above confirmed that there were two characteristic bands in the ECD spectra of 1″*S*,2″*S*-maackiazin (**10**)—a positive band at λ ≈ 200 nm and the high-frequency part of a negative band doublet in the region 195 ≤ λ ≤ 220 nm. No other bands were characteristic due to the effective averaging of their intensities over LAM processes, which gave approximately zero. The comparison of the theoretical statistically averaged spectra, calculated for 1″*S*,2″*S* and 1″*R*,2″*R* stereoisomers of maackiazin (**10**) with the experimental ECD spectrum of **10**, thus, must be restricted to the analysis of their shapes at λ ≤ 250. A good correspondence between calculated and experimental spectra was observed only for 1″*S*,2″*S* stereoisomer of **10** (Figure 2).

The absolute configuration of scirpusin A (**13**) was determined using the same approach. To distinguish between 7*R*,8*R*- and 7*S*,8*R*-scirpusin A (**13**) the J (H7,H8) vicinal constants were calculated and compared to the experimental value.

The performed conformational analysis of scirpusin A (**13**) showed that the potential energy profile for inversion of the dihydrofuran cycle in **13** depended on the configurations of its asymmetric centers. Thus, 7*S*,8*R*-scirpusin A (**13**) had only one stable conformation of the dihydrofuran cycle, whereas 7*R*,8*R*-scirpusin A (**13**)—two conformations. This difference in the topography of the potential energy surfaces led to differences in the predicted values of the vicinal spin–spin coupling constant. The value of J (H7,H8) was 9.2–9.4 Hz for 7*S*,8*R*-scirpusin A (**13**) and 6.3 Hz for 7*R*,8*R*-scirpusin A (**13**). The experimental value of 5.3 Hz correlated well enough with the last calculated value (Appendix A). Thus, the relative configuration of **13** was determined as 7*R*,8*R* or 7*S*,8*S*. Then, we calculated ECD spectra for these stereoisomers. We found that scirpusin A (**13**) was structurally flexible. Thus, the rotation of the *p*-vinylphenol substituent around C8′-C9′ was dynamically coupled to the inversion of the dihydrofuran cycle. Both these processes led to variations in the shape of the ECD spectrum, but the sign of the Cotton effect in the characteristic region at λ ≤ 240 nm did not change. The position of the band at λ ≈ 200 nm in experimental UV spectrum was chosen to be the reference for the determination of the UV shift Δλ = +17 nm (Appendix A).

The calculated ECD spectra for 7*R*,8*R*-scirpusin A (**13**) and 7*S*,8*S*-scirpusin A (**13**) compared to the experimental spectrum for **13** are presented in Figure 3. A good correspondence between the calculated ECD spectrum for 7*R*,8*R*-scirpusin A (**13**) and the experimental spectrum confirmed the 7*R*,8*R* configuration of **13**.

The positive Cotton effect at 331 nm in the CD spectrum of liquiritigenin (**6**) confirmed its 2*S* configuration [23].

### 2.3. In Vitro Model of 6-OHDA Induced Neurotoxicity (Cell Viability Test)

Polyphenolic compounds **1**–**13** did not possess cytotoxic activity against Neuro-2a cells at a concentration up to 100 µM.

Neuro-2a cells were incubated with the studied polyphenolic compounds for 1 h. Then, 100 µM of 6-hydroxydopamine 6-OHDA were added. We determined the percentage of living Neuro-2a cells after treatment with 6-OHDA (Figure 4a–d). Liquiritigenin (**6**) and scirpusin A (**13**) at a concentration of 10 µM, as well as Maksar^®^ (10 µg/mL), significantly increased the viability of 6-OHDA-treated Neuro-2a cells by 12%, 9%, and 11%, respectively (Figure 4b,d), whereas isoflavonoids **4**, **5**, **7**, **8**, maackiazin (**10**) and piceatannol (**11**) showed a weak effect on cell viability. Compounds **8** and **12** increased the viability of Neuro-2a cells at a concentration of 1 µM. Maackin (**9**) did not show significant activity in this test (Figure 4c), which was in accordance with the data obtained using a paraquat (PQ)-induced neurotoxicity model [22]. Resveratrol also did not cause any increase in the viability of 6-OHDA-treated Neuro-2a cells.

### 2.4. Reactive Oxygen Species (ROS) Analysis in 6-OHDA-Treated Cells

We studied the effect of polyphenolic compounds set on ROS levels in 6-OHDA-treated Neuro-2a cells (Figure 5a–d). Scirpusin A (**13**), maackiasin (**10**), and liquiritigenin (**6**) possessed high activity in this test and at a concentration of 1 µM/mL they reduced ROS levels in 6-OHDA-treated Neuro-2a cells by 43%, 20%, and 15%, respectively (Figure 5a–d). Liquiritigenin (**6**) showed a dose-dependent effect (Figure 5b). Resveratrol (**1**), though it did not increase cell viability, at a concentration of 1 µM/mL it decreased ROS levels in 6-OHDA-treated cells by approximately 15%.

### 2.5. Mitochondrial Membrane Potential (MMP) Detection

We determined the effect of polyphenolic compounds from *M. amurensis* on 6-OHDA-induced mitochondrial dysfunction in Neuro-2a cells (Figure 6a,b). The tetramethylrhodamine methyl (TMRM) fluorescence decreased by 14% after the treatment of Neuro-2a cells with 6-OHDA, which indicated that 6-OHDA caused depolarization of the mitochondrial membrane. We observed a significant effect of increasing mitochondrial membrane potential by 8–12% when Neuro-2a cells were incubated with resveratrol (**1**), piceatannol (**11**), liquiritigenin (**6**), medicarpin (**7**), and maackin (**9**) at a concentration of 10 μM (Figure 6a,b). Retusin (**3**) at a concentration of 10 μM showed weak activity in this test.

### 2.6. Cell Viability and Reactive Oxygen Species (ROS) Analysis in PQ-Treated Cells

We previously studied the effect of stilbens **9**, **10**, **11**, and **13** on cell viability and ROS level in PQ-treated Neuro-2a cells. Here, we evaluated the neuroprotective potential against the toxic effect of PQ of stilbene **1**, flavonoid **6** and isoflavonoids **2**–**5**, **7**, **12** (Figure 7a–d). Resveratrol (**1**) and liquiritigenin (**6**) at a concentration of 10 µM demonstrated considerable activity and increased the viability of PQ-treated cells by 17% and 15%, respectively (Figure 7a–c). Liquiritigenin (**6**) also effectively reduced the ROS level in PQ-treated Neuro-2a cells (Figure 7d). Isoflavonoids genistein (**2**), medicarpin (**7**), and maackiain (**4**) at this concentration showed a much less significant effect and increased the cell viability by 5%, 7% and 2%, respectively (Figure 7a–c).

### 2.7. Apoptosis

We investigated the effect of polyphenolic compounds from *M. amurensis* on apoptosis profile in 6-OHDA-treated Neuro-2a cells by flow cytometry using fluorescent annexin V conjugate as a fluorescent dye [24]. We observed significant changes in apoptosis profile in Neuro-2a cells after 6-OHDA-treatment. The percentages of early apoptotic and late apoptotic cells increased after 6-OHDA-treatment from 4.59 to 42.62% and from 4.3 to 12.15%, respectively (Table 1). Polyphenols **3**, **6** and **13** diminished the neurotoxic effect of 6-OHDA-treatment. These compounds considerably increased the percentage of live Neuro-2a cells and decreased the number of early apoptotic cells (Table 1, Figure 8).

### 2.8. Virucidal Activity of Polyphenolic Compounds from M. amurensis (CPE Inhibition Assay)

We determined the cytotoxicity of polyphenolic compounds **1**–**14** against Vero cells using the MTT assay. The 50% cytotoxic concentrations (CC_50_) of the studied compounds were in the range of 120–300 μg/mL (Table 2).

We previously reported that stilbenes **9**–**11**, **13**, and **14** possessed anti-HSV-1 activity when added to Vero cells simultaneously with the virus [22]. This method of exposure of the virus and cells to polyphenols was used to study the ability of the compounds to prevent the penetration of virus to cells. Here, we investigated the ability of polyphenols **1**–**14** to directly interact with the viral particles (HSV-1 was pretreated with the studied compounds) [25,26,27].

The antiviral activity of polyphenols from *M. amurensis* against HSV-1 was assessed using the cytopathic effect inhibition (CPE) assay. To study the inhibitory effect of polyphenolic compounds directly on the HSV-1 virus (direct virucidal activity), the monolayer of Vero cells grown on 96-well plates (1 × 10^4^ cells/well) was infected with the virus (100 TCD_50_/mL), pretreated with different concentrations of polyphenolic compounds. The antiviral activity of compounds **1**–**14** was studied at concentrations below CC_50_ values (Table 2, Appendix A). Maackin (**9**) and scirpusin A (**13**) inhibited viral replication more effectively than the other studied polyphenols. (IC_50_ values were 2.4 and 2.6 μg/mL, and SI values were 112.3 and 76.3, respectively). Polyphenolic compounds **1**, **7**, **10**, **11**, **14** also showed significant antiviral activity with IC_50_ values ranging from 6.7 to 9.6 μg/mL. Genistein (**2**) and maackiain (**4**) showed moderate activity with SI values of 8.0 and 5.4 and IC_50_ values of 16.7 and 28.3 μg/mL, respectively (Table 2). Polyphenols **3**, **5**, **6**, **8** and **12** did not show virucidal activity against HSV-1.

### 2.9. Virucidal Activity of Polyphenolic Compounds from M. amurensis (RT-PCR Assay)

The anti-HSV-1 activity of polyphenolic compounds from *M. amurensis* was also studied using the real-time PCR (RT-PCR) technique (Table 3, Appendix A). Vero cells grown on 96-well plates were infected with the virus (100 TCID_50_/mL), pretreated with polyphenolic compounds at concentrations 0.1, 1 and 10 μg/mL. After 72 h of cell incubation, viral DNA was extracted from the supernatants, and the relative level of HSV-1 DNA was determined using RT-PCR. The effect of polyphenols on the relative level of viral DNA was assessed using the 2^−∆Ct^ method and reported as fold reduction compared to virus control.

We found that several polyphenolic compounds from *M. amurensis* at a concentration of 10 µg/mL significantly inhibited the replication of HSV-1 (Table 3). Maackin (**9**) and scirpusin A (**13**) caused a maximum reduction in viral DNA (4.6 log10 and 3.9 log10, respectively; *p* < 0.05) at this concentration. Compounds **10**, **11** and **14** also significantly reduced the level of viral DNA. Compounds **1**, **2**, **4**, and **7** possessed moderate anti-HSV-1 activity. Only compounds **9**, **11, 13** inhibited the replication of HSV-1 at a concentration 1 µg/mL (Table 3). However, we did not observe any significant reduction in viral load when HSV-1 was pretreated with polyphenols at a concentration of 0.1 μg/mL. Polyphenols **3**, **5**, **6**, **8** and **12** did not demonstrate virucidal activity against HSV-1.

### 2.10. The Effect of Polyphenols on the Early Stages of HSV-1 Infection

The study of the virucidal activity of polyphenolic compounds from *M. amurensis* revealed that maackin (**9**) and scirpusin A (**13**) exhibited the highest inhibitory activity against the HSV-1 virus among the tested compounds (Table 2 and Table 3). In order to study the mechanism of action of these compounds, two additional schemes of treatment of the virus and Vero cells with polyphenols were applied: the compounds were added before virus infection (pretreatment of cells) and after penetration of the virus into host cells (treatment of infected cells). The fourth method of exposure where the virus and polyphenols were added to cells simultaneously (simultaneous treatment) was previously described by us in [22]. The anti-HSV-1 activity of polyphenols was assessed using the cytopathic effect inhibition (CPE) assay. The obtained results are presented in Table 4.

The treatment of Vero cells with maackin (**9**) and scirpusin A (**13**) before infection (pretreatment of cells) had a weak effect on HSV-1 replication (IC_50_ 70.3 and 47.2 µg/mL, SI 3.8 and 4.2, respectively) (Table 4). A moderate inhibition of virus replication was observed after simultaneous treatment of cells with the virus and the tested polyphenols [22]. The application of polyphenols after virus adsorption and penetration to cells (treatment of infected cells) also had a moderate effect on HSV-1 replication (Table 4).

Thus, we showed that the tested polyphenols, especially maackin (**9**) and scirpusin A (**13**), possessed significant anti-HSV-1 activity mainly due to their direct virucidal effect. These compounds also inhibited virus–cell interactions when added simultaneously with the initiation of viral infection or even after virus adsorption and penetration to cells.

## 3. Discussion

Here, we continued our investigation of anti-HSV-1 activity and neuroprotective potential of isoflavonoids and stilbenes present in Maksar^®^.

The neuroprotective potential of polyphenolic compounds from *M. amurensis* heartwood was studied using models of 6-OHDA-and PQ-induced neurotoxicity [28,29]. Being a structural analogue of dopamine, 6-OHDA is an endogenous neurotoxin. It selectively penetrates dopaminergic and noradrenergic neurons, accumulates in the cytosol, and changes into dihydrophenylacetic acid or oxidizes to form hydrogen peroxide and para-quinone, which leads to ROS formation and oxidative stress in neurons, followed by cell death [28]. A dimeric stilbene scirpusin A (**13**) and a flavonoid liquiritigenin (**6**) were shown to be the most active compounds among the tested polyphenols preventing ROS formation in neuronal cells. These compounds significantly increased the viability of 6-OHDA-treated Neuro-2a cells. This effect was due to the ability of these polyphenols to reduce the level of intracellular ROS (Figure 4 and Figure 5). Liquiritigenin (**6**) also increased mitochondrial membrane potential (Figure 6). Although resveratrol (**1**) previously demonstrated a protective effect against several type of insults that have shown a correlation to Parkinson’s disease (PD) pathogenesis, it did not increase the viability of 6-OHDA-treated Neuro-2a cells in our study [30]. However, resveratrol (**1**) caused a significant decrease in intracellular ROS level and an increase in mitochondrial membrane potential in 6-OHDA-treated Neuro-2a cells. Maackiazin (**10**) had weak effect on cell viability, but considerably reduced ROS level in 6-OHDA-treated Neuro-2a cells. Compound **3** considerably inhibited ROS formation only at a concentration of 0.1 µM, which can be due to the possible prooxidant effect of this compound. Notably, the mechanism of action of polyphenolic compounds is complex and multitarget. This may include possible interactions with enzymes or receptors in a cell, which result a “bell-shaped” or reverse dependence of the dose-response for some polyphenolic compounds.

In contrast to 6-OHDA, PQ is an exogenous neurotoxin. PQ effects the redox cycle and increases ROS production [28]. Here, we evaluated the neuroprotective potential against the toxic effect of PQ of stilbene resveratrol (**1**), flavonoid liquiritigenin (**6**) and isoflavonoids, genistein (**2**), retusin (**3**), maackiain (**4**), tectorigenin (**5**), medicarpin (**7**), and (±)-3-hydroxyvestitone (**12**) (Figure 7).

Resveratrol (**1**) and liquiritigenin (**6**) at a concentration of 10 µM demonstrated considerable activity and increased the viability of PQ-treated cells by 17% and 15%, respectively. These compounds had significant effect even at a concentration of 1 µM (Figure 7a–c). Liquiritigenin (**6**) also effectively reduced the ROS level in PQ-treated Neuro-2a cells (Figure 7d). Isoflavonoids genistein (**2**), medicarpin (**7**), and maackiain (**4**) at this concentration showed weak effect on cell viability (Figure 7a–c).

Here, we also investigated the effect of polyphenolic compounds from *M. amurensis* on apoptosis profile in 6-OHDA-treated Neuro-2a cells by flow cytometry. We observed significant changes in apoptosis profile in Neuro-2a cells after 6-OHDA-treatment. Polyphenols **3**, **6** and **13** diminished the neurotoxic effect of 6-OHDA. These compounds considerably increased the percentage of live Neuro-2a cells and decreased the number of early apoptotic cells (Table 1, Figure 8).

The CPE inhibition assay and RT-PCR showed that dimeric stilbens with *trans*-double bond maackin (**9**) and scirpusin A (**13**) possessed the highest anti-HSV-1 activity among polyphenols **1**–**14** (Table 2 and Table 3). The study of the effect of polyphenols **9** and **13** on the early stages of HSV-1 infection revealed that the main mechanism of their antiviral activity was direct interaction with the virus (virucidal activity) (Table 4). This activity may be due to the ability of these polyphenols to interact with the surface glycoproteins of HSV-1 or lipid components of the viral envelope.

Another mechanism of the antiviral action of polyphenols with pronounced antioxidant properties is the reduction in HSV-1-induced ROS production in the cells [31]. In our study, stilbenes **1**, **9**–**11**, **13**, **14** demonstrated much more pronounced anti-HSV-1 activity compared to isoflavonoids (**2**–**5**, **7**, **8**, **12**) (Table 2 and Table 3), which may be due to the fact that stilbenes from *M. amurensis* possessed higher antiradical and antioxidant activity than isoflavonoids [6,22]. Notably, promising anti-HSV-1 activity of maackin (**9**) and scirpusin A (**13**) correlated with high antiradical and antioxidant activity of these stilbenes.

## 4. Materials and Methods

### 4.1. Plant Material

We provided the information on how *M. amurensis* was collected in our previous report [22].

### 4.2. Extraction and Isolation

We extracted *M. amurensis* heartwood (500 g) twice with a mixture of CHCl_3_–EtOH at a ratio of 3:1 (*v*/*v*) for 3 h (60 °C) as previously described [22]. The air-dried extract (15 g) was applied to a polyamide column (50–160 µm, Sigma-Aldrich, St. Louis, MI, USA) and eluted with hexane–CHCl_3_ and CHCl_3_–EtOH solution systems to obtain fractions 1–8 and 9–18, respectively. We gradually increased the portions of CHCl_3_ and EtOH in the solution systems (hexane/CHCl_3_, *v*/*v*: 1:0, 10:0, 8:1, 5:1, 2:1, 1:1, 1:2, CHCl_3_; CHCl_3_/EtOH, *v*/*v*: 1:0, 100:1, 50:1, 40:1, 30:1, 20:1, 10:1, 5:1, 2:1). The fractions containing polyphenolic compounds according to HPLC data were selected for further purification.

Fraction 9 (630 mg) was subsequently chromatographed on a silica gel column (40–63 µm, Sigma-Aldrich, St. Louis, MI, USA) twice to obtain pure compounds **3** (6.3 mg) and **4** (5.2 mg). Fraction 10 (895 mg) was also chromatographed on a silica gel column twice to obtain individual compounds **2** (25.8 mg), **5** (7.8 mg), **7** (8.3 mg), **8** (33.6 mg), and **12** (3.7 mg). Compounds **6** (5.8 mg) and **10** (7.3 mg) were isolated from fraction 11 (654 mg) by silica gel chromatography. Fraction 13 (721 mg) contained compounds **1** (10.1 mg), **9** (12.4 mg), **11** (17.2 mg), **13** (14.3 mg), and **14** (3.1 mg). These polyphenols were separated on a silica gel column and subsequently purified on a C-18 column (YMC gel ODS-A 75 μm, Kyoto, Japan).

### 4.3. General Experimental Procedures

The CD spectra were measured on a Chirascan-plus Quick Start CD Spectrometer (Applied Photophysics Limited, Leatherhead, UK) (acetonitrile, 20 °C). We recorded the ^1^H NMR spectra in acetone on an NMR Bruker AVANCE III DRX-700 instrument (Bruker, Karlsruhe, Germany).

### 4.4. HPLC–PDA–MS

We analyzed the fractions and individual polyphenolic compounds using a HPLC–PDA–MS system consisting of a CBM-20A system controller (Shimadzu, Kyoto, Japan), two LC-20AD pumps (Shimadzu, Kyoto, Japan), a DGU-20A3 degasser (Shimadzu, Kyoto, Japan), SIL-20A autosampler (Shimadzu, Kyoto, Japan), an SPD-M20A UV–VIS photodiode array detector (Shimadzu, Kyoto, Japan), and a LCMS-2020 mass detector (Shimadzu, Kyoto, Japan). The compounds were analyzed on a Discovery XR-ODS column (15 × 2.1 mm i.d.; 3.0 μm particle size; Supelco Analytical, Bellefonte, PA, USA) at a flow rate of 0.3 mL/min. The column was thermostated at 40 °C. The mobile phase consisted of 1% aqueous acetic acid (A) and acetonitrile containing 1% of acetic acid (B). We programmed the following gradient steps: 25–35% B (0–6 min), 35–60% B (6–11 min), 60–90% B (11–14 min), 90–25% B (14–16 min), 25% B (16–20 min). The injection volume was 2 μL. The MS settings were as follows: electrospray ionization (ESI), negative and positive ion modes, 150–800 *m*/*z* scans, N_2_ drying gas (10 L/min), nebulizer gas flow 1.5 L/min, interface voltage 3.5 kV, and detector voltage 1.2 kV. We acquired and processed the data using Shimadzu LCMS Solution software (v. 5.42, Kyoto, Japan).

### 4.5. Neuro-2a Cell Line and Culture Condition

We acquired the murine neuroblastoma cell line Neuro-2a from American Type Culture Collection (ATCC^®^ CCL-131, Manassas, VA, USA). We cultured the cells in DMEM supplied with 10% fetal bovine serum (Biolot, St. Petersburg, Russia) and 1% penicillin/streptomycin (Biolot, St. Petersburg, Russia) as previously described [22].

We performed the MTT assay to evaluate the cytotoxic activity of polyphenolic compounds against Neuro-2a as previously described [22].

### 4.6. In Vitro Model of 6-OHDA and PQ-Induced Neurotoxicity (Cell Viability Test)

After 24 h of adhesion, Neuro-2a cells (5 × 10^4^ cells/mL) were treated with polyphenolic compounds at concentrations of 0.1, 1 and 10 μM for 1 h. Then, 100 µM of 6-OHDA or 1 mM of PQ (Sigma-Aldrich, St. Louis, MO, USA) were added. Cells incubated without inductors or with inductors were used as positive and negative control, respectively. Cell viability was measured after 24 h using the MTT assay.

### 4.7. Reactive Oxygen Species (ROS) Analysis in 6-OHDA- and PQ-Treated Cells

After 24 h of adhesion, Neuro-2a cells (5 × 10^4^ cells/mL) were incubated with polyphenolic compounds at concentrations of 0.1, 1, and 10 µM for 1 h. Then, 120 µM of 6-OHDA or 1 mM of PQ were added to each well, and cells were incubated for 1 h. To study the ROS formation, 20 µL of 2,7-dichlorodihydrofluorescein diacetate solution (H2DCF-DA, Sigma-Aldrich, St. Louis, MO, USA) were added to each well, so that the final concentration was 10 µM. Then, the microplate was incubated for 30 min at 37 °C.

### 4.8. Mitochondrial Membrane Potential (MMP) Detection

The cells were incubated for 1 h in a 96-well plate (1 × 10^4^ cells/well) with polyphenolic compounds (0.1, 1 and 10 µM). Then, 6-OHDA (120 µM) was added, and cell suspension was incubated for 1 h. Cells incubated without 6-OHDA and compounds and with 6-OHDA alone were used as positive and negative control, respectively. The tetramethylrhodamine methyl (TMRM) (Sigma-Aldrich, St. Louis, MO, USA) solution (500 nM) was added to each well and cells were incubated for 30 min at 37 °C. The intensity of fluorescence was measured using a PHERAstar FSplate reader (BMG Labtech, Ortenberg, Germany) at λ_ex_ = 540 nm and λ_em_ = 590 nm. The data were processed by MARS Data Analysis v. 3.01R2 (BMG Labtech, Ortenberg, Germany).

### 4.9. Apoptosis

The cells were incubated for 1 h in a 6-well plate (5 × 10^4^ cells/mL) with polyphenolic compounds (10 µM). Then, 6-OHDA (100 µM) was added, and cell suspension was incubated for 24 h. After incubation, the maintenance media was collected, and the cells were washed by cold PBS twice and incubated with trypsin-EDTA solution. The cell suspension was washed with cold PBS twice again, centrifugated and then used for apoptosis detection by Muse^®^ Annexin V & Dead Cell Kit in accordance with the manufacturer’s instructions (Luminex, Austin, TX, USA). The fluorescence was measured with Muse^®^ Cell Analyzer (Luminex, Austin, TX, USA) and the data were processed by Muse 1.5 Analysis software (Luminex, Austin, TX, USA).

### 4.10. HSV-1 Virus and Vero Cell Culture

The HSV-1 strain L2 obtained from N.F. Gamaleya Federal Research Centre for Epidemiology and Microbiology (Moscow, Russia) was grown in Vero cells as previously described [22].

We prepared the stock solutions of the tested compounds (10 mg/mL) in DMSO (Sigma, St. Louis, MO, USA). For the MTT assay and further studies of anti-HSV-1 activity, we diluted the stock solutions with DMEM so that the final concentration of DMSO was 0.5%.

### 4.11. Cytotoxicity of the Tested Compounds against Vero Cells

We used MTT assay to determine the cytotoxic effect of the tested compounds against Vero cells. We performed the MTT assay as previously described [22].

### 4.12. Virucidal Activity of Polyphenolic Compounds from M. amurensis

We determined the direct inhibitory effect of polyphenolic compounds from *M. amurensis* heartwood on HSV-1 (virucidal effect). We mixed the virus at an infectious dose (100 TCID_50_/mL) with polyphenolic compounds (from 0.1 to 100 µg/mL) at a ratio of 1:1 (*v*/*v*) and incubated for an hour at 37 °C. Then, we infected the monolayer of Vero cells grown on 96-well plates (1 × 10^4^ cells/well) with HSV-1 pretreated with the studied compounds. After 1 h adsorption at 37 °C, the cells were washed with phosphate-buffered saline (PBS) and overlaid by the maintenance medium with 1% FBS. The plates were incubated for 72 h at 37 °C in a CO_2_-incubator. The cytopathic effect (CPE) inhibition assay and real-time reverse transcription-PCR (RT-PCR) assay were used to evaluate virucidal effect of the tested compounds.

#### 4.12.1. Cytopathic Effect (CPE) Inhibition Assay

We previously described this method in detail in [22]. We performed the MTT assay to evaluate the antiviral activity of polyphenolic compounds. The viral inhibition rate (IR, %) was calculated according to Equation (1) [32]:(1)$$\mathrm{IR},\;\%=\frac{A_{tv}-A_{cv}}{A_{cd}-A_{cv}}\times 100,$$
where *A_tv_* is the absorbance of cells infected with HSV-1 pretreated with the studied compounds,

*A_cv_* is the absorbance of the virus-infected cells, but not treated with the studied compounds,

*A_cd_* is the absorbance of control cells (untreated and non-infected).

We calculated 50% inhibitory concentration (the concentration of the compound that reduced the virus-induced CPE by 50%, IC_50_) using a regression analysis of the dose–response curve [33,34]. We calculated the selectivity index (SI) as the ratio of CC_50_ to IC_50_. Experiments were repeated three times.

#### 4.12.2. Real-Time Polymerase Chain Reaction (RT-PCR) Assay

We performed the RT-PCR assay as we previously described in detail in [22]. Briefly, HSV-1 DNA was extracted after incubation from cell supernatants by using the DNA-sorb-AM (K1-12-100-CE) nucleic acid extraction kit (AmpliSens^®^_,_ Moscow, Russia) according to the manufacturer′s instructions. We treated the supernatants with a lysis solution that contained guanidine chloride (chaotropic agent) in the presence of sorbent (silica particles). As the elution solution was added, the DNA was adsorbed on silica particles and separated from the sorbent particles by centrifugation.

We detected the DNA of HSV-1 in the obtained samples using the AmpliSens^®^ HSV I, II-FRT-100F PCR kit (AmpliSens^®^_,_ Moscow, Russia) on a real-time PCR instrument Rotor-Gene Q (Qiagen, Hilden, Nordrhein-Westfalen, Germany) according to the manufacturer′s instructions. The PCR reaction conditions were the following: initial denaturation at 95 °C for 15 min; then 5 cycles at 95 °C for 5 s, at 60 °C for 20 s, at 72 °C for 15 s; then 40 cycles at 95 °C for 5 s, at 60 °C for 20 s, and at 72 °C for 15 s. A negative sample was used as the amplification control for each run. The threshold cycle number *C_t_*, was measured as the PCR cycle, where the amount of the amplified target reached the threshold value; the *C_t_* values ≥ 37 indicated the absence of the HSV-1 DNA in the samples. We assessed the anti-HSV-1 activity of the studied compounds by the reduction in the viral DNA levels using the 2^−ΔCt^ method [35] according to Equation (2):(2)$$\Delta C_{t}=C_{tcv}-C_{ttv}$$
where *C_t_* is the threshold cycle number measured as the PCR cycle, where the amount of the amplified target reached the threshold value,

*C_ttv_* is the average *C_t_* value for the infected samples (the virus was pretreated with polyphenolic compounds),

*C_tcv_* is the average *C_t_* value for virus control.

### 4.13. The Effect of Polyphenols on the Early Stages of HSV-1 Infection

In order to more deeply study the anti-HSV-1 effect of polyphenolic compounds that exhibited high virucidal activity we applied several additional schemes.

#### 4.13.1. Pretreatment of Cells with Polyphenolic Compounds

The monolayer of cells was pretreated with different concentrations of the studied compounds for 2 h at 37 °C. After washing, the cells were infected with 100 TCID_50_/mL of the virus at 37 °C for 1 h. Then, unabsorbed virus was removed by washing with PBS (Sigma, Saint-Louis, MO, USA), and cells were incubated in the maintenance medium until CPE appeared.

#### 4.13.2. Treatment of Virus-Infected Cells with Polyphenols

The monolayer of cells was infected with the virus (100 TCID_50_/mL) at 37 °C for 1 h, and then the cells were washed with PBS (Sigma, Saint-Louis, MO, USA) and treated with different concentrations of the studied compounds and incubated until CPE appeared.

After the incubation, MTT assay was performed and viral inhibition rate (IR), 50% inhibitory concentration (IC_50_) and selectivity index (SI) of these polyphenolic compounds were calculated.

### 4.14. Statistical Analysis

The statistical analysis was performed using Statistica 10.0 software (StatSoft, Inc., Tulsa, OK, USA). CC_50_ and IC_50_ values were calculated using linear regression analysis of the dose-response curve. Results are presented as the mean values and standard deviations of three or more independent experiments. Comparison of differences between the indicators of the control and experimental groups was carried out using the Wilcoxon test for related samples. The differences were considered statistically significant if the *p*-value did not exceed 0.05.

## 5. Conclusions

The polyphenolic compounds contribute to the antiviral and neuroprotective activity of Maksar^®^. Stilbens scirpucin A (**13**) and maackin (**9**) possessed the highest anti-HSV-1 activity among the tested polyphenols. Notably, these stilbenes were more active than resveratrol, which was previously shown to possess antiherpetic properties [36,37]. Liquiritigenin (**6**) and scirpucin A (**13**) were mainly responsible for the neuroprotective potential of Maksar^®^. Scirpusin A (**13**) was the most promising compound possessing both anti-HSV-1 activity and neuroprotective potential.

Thus, Maksar^®^ can be a perspective drug for the treatment of herpetic infections and neuronal disorders.

## Figures and Tables

**Figure 1 ijms-25-04142-f001:**
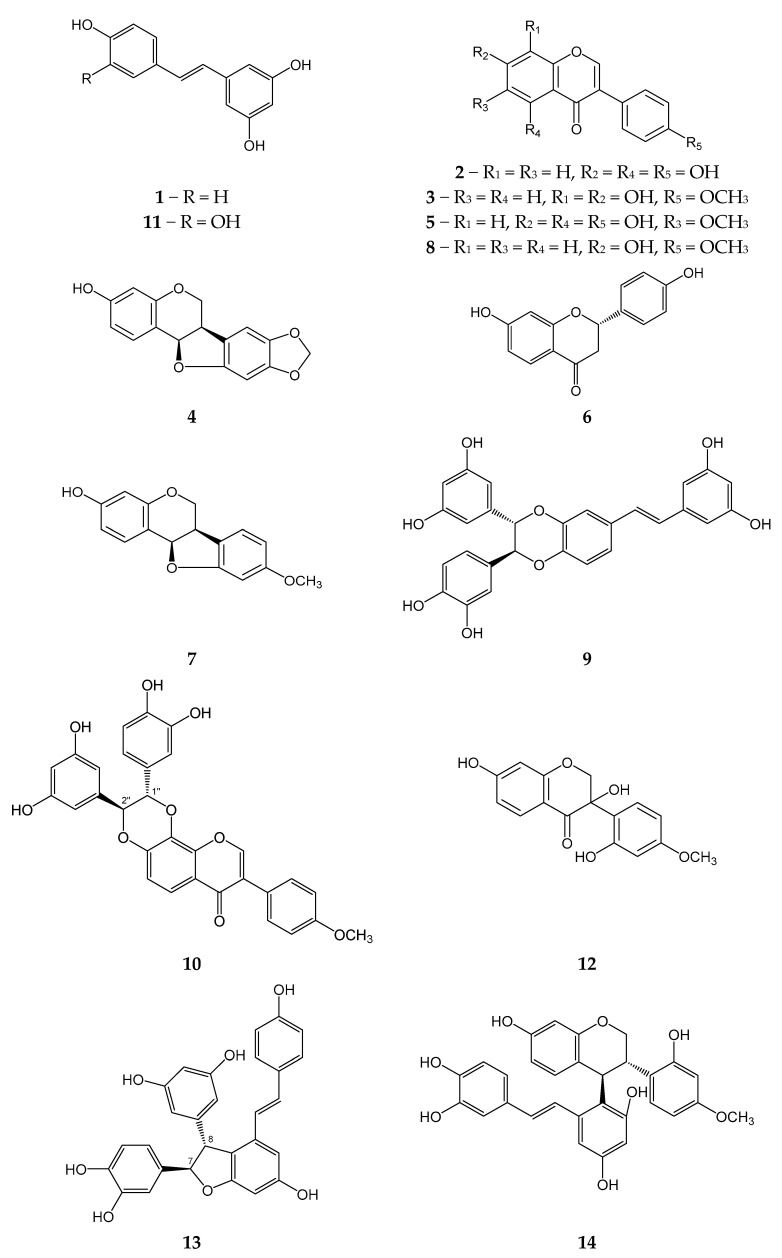
Polyphenolic compounds from *M. amurensis* heartwood.

**Figure 2 ijms-25-04142-f002:**
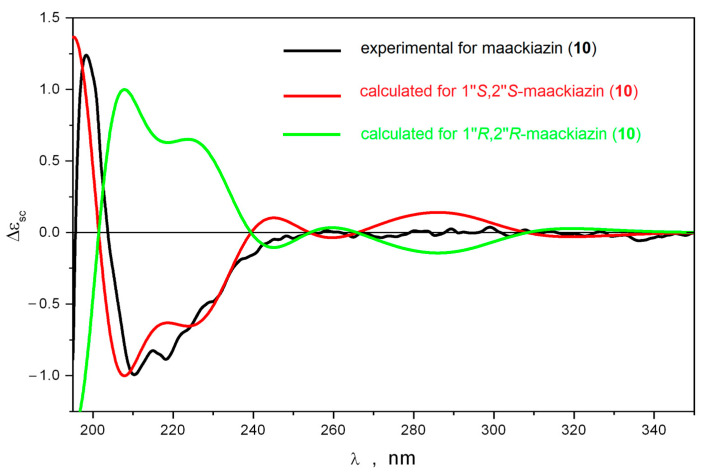
The comparison of the experimental and calculated ECD spectra for maackiazin (**10**).

**Figure 3 ijms-25-04142-f003:**
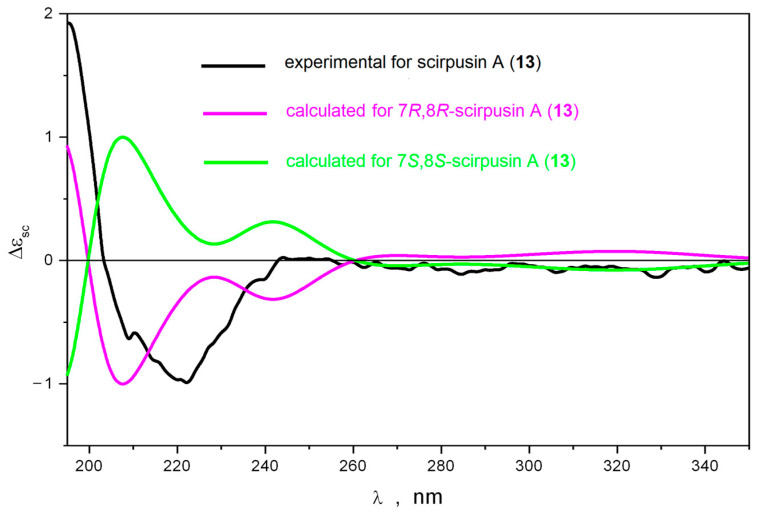
The comparison of experimental and calculated ECD spectra for scirpusin A (**13**).

**Figure 4 ijms-25-04142-f004:**
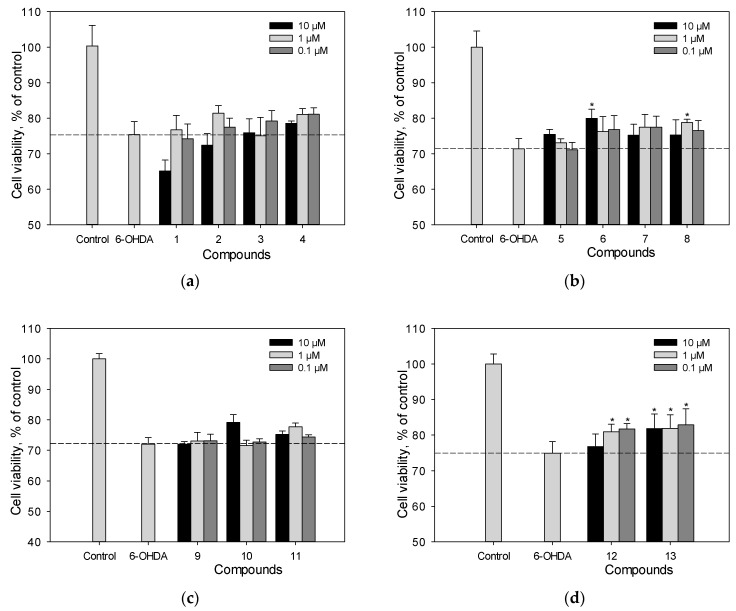
The effect of polyphenolic compounds isolated from *M. amurensis* heartwood on the viability of Neuro-2a cells treated with 6-OHDA (100 μM) (**a**–**d**). The viability of Neuro-2a cells treated with various compounds and 6-OHDA was measured by the MTT assay. Polyphenolic compounds isolated from *M. amurensis* heartwood were used at concentrations 10, 1, 0.1 μM. The data are presented as means ± SEM of three independent replicates. (*) indicate *p* < 0.05 versus 6-OHDA-treated cells.

**Figure 5 ijms-25-04142-f005:**
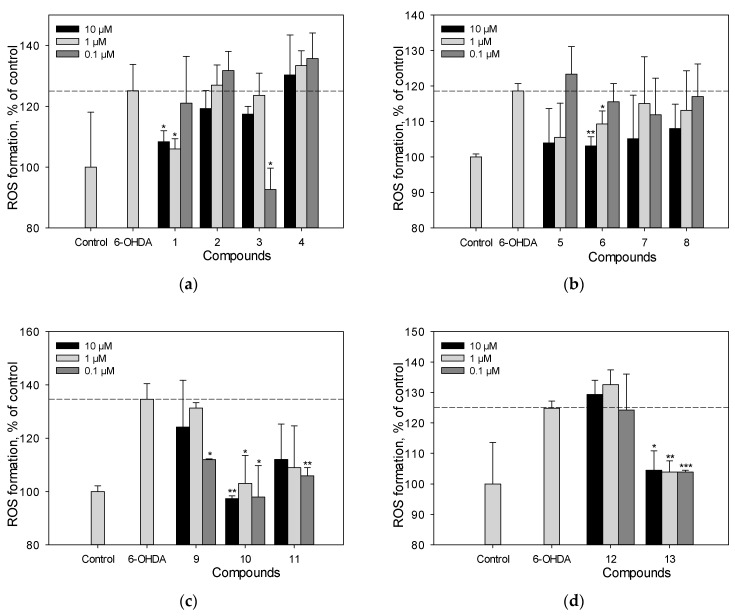
The effect of polyphenolic compounds isolated from *M. amurensis* heartwood on ROS levels in Neuro-2a cells treated with 6-OHDA (120 μM) (**a**–**d**). Polyphenolic compounds isolated from *M. amurensis* heartwood were used at concentrations 10, 1, 0.1 μM. The data are presented as means ± SEM of three independent replicates. (*), (**), and (***) indicate, respectively, *p* < 0.05, *p* < 0.005, and *p* < 0.001 versus 6-OHDA-treated cells.

**Figure 6 ijms-25-04142-f006:**
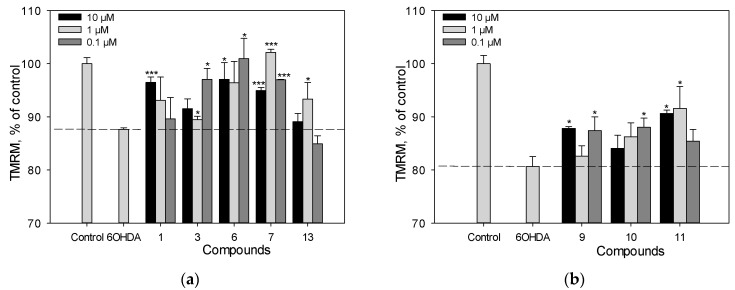
The effect of polyphenolic compounds isolated from *M. amurensis* heartwood on mitochondrial membrane potential in Neuro-2a cells treated with 6-OHDA (120 μM) (**a**,**b**). Polyphenolic compounds isolated from *M. amurensis* heartwood were used at concentrations 10, 1, 0.1 μM. The data are presented as means ± SEM of three independent replicates. (*) and (***) indicate, respectively, *p* < 0.05 and *p* < 0.001 versus 6-OHDA-treated cells.

**Figure 7 ijms-25-04142-f007:**
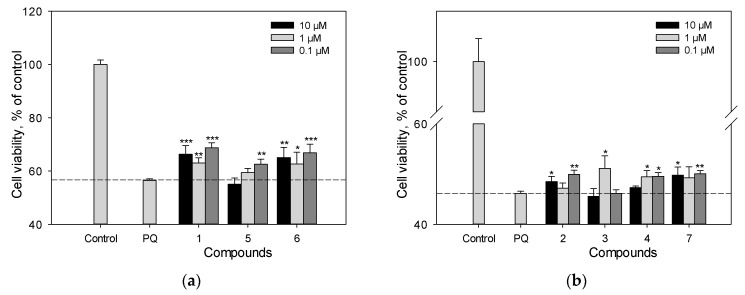
The effect of polyphenolic compounds isolated from *M. amurensis* heartwood on cell viability (**a**–**c**) and ROS levels (**d**) in Neuro-2a cells treated with PQ (1 mM). Polyphenolic compounds isolated from *M. amurensis* heartwood were used at concentrations 10, 1, 0.1 μM. The data are presented as means ± SEM of three independent replicates. (*), (**), and (***) indicate, respectively, *p* < 0.05, *p* < 0.005, and *p* < 0.001 versus PQ-treated cells.

**Figure 8 ijms-25-04142-f008:**
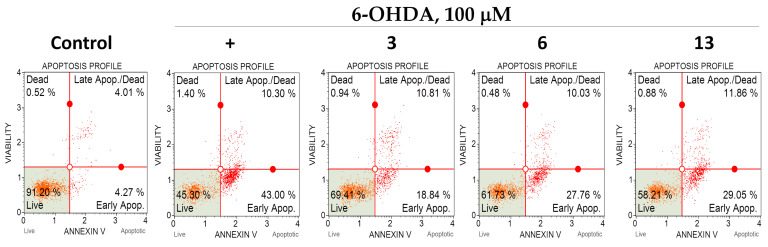
The effect of polyphenolic compounds isolated from *M. amurensis* heartwood on apoptosis profile in 6-OHDA-treated Neuro-2a cells.

**Table 1 ijms-25-04142-t001:** The effect of polyphenolic compounds from *M. amurensis* heartwood on apoptosis profile in 6-OHDA-treated Neuro-2a cells.

	Live	Early apop.	Late apop./Dead	Dead
Control	90.53 ± 0.68	4.59 ± 0.32	4.3 ± 0.29	0.63 ± 0.05
6-OHDA	44.14 ± 1.34	42.62 ± 2.3	12.15 ± 1.34	1.11 ± 0.37
**3**	67.89 ± 1.52 *	19.63 ± 0.79 *	11.31 ± 0.49	1.18 ± 0.24
**6**	61.83 ± 0.09 *	27.67 ± 1.09 *	10.89 ± 0.86	0.61 ± 0.13
**13**	56.90 ± 1.32 *	30.18 ± 1.13 *	12.2 ± 0.34	0.73 ± 0.15

* *p* ≤ 0.05 compared to 6-OHDA-treated Neuro-2a cells.

**Table 2 ijms-25-04142-t002:** Virucidal activity of polyphenolic compounds from *M. amurensis* heartwood (CPE assay) against HSV-1. Values are given as the means ± SD of three or more independent experiments. Acyclovir^®^ was used as the positive control.

Compounds	CC_50_		IC_50_		SI
	µg/mL	µM	µg/mL	µM	
**1**	123.0 ± 14.8	539.5 ± 64.9	9.6 ± 1.4	42.1 ± 6.1	12.8 ± 1.9
**2**	133.8 ± 20.1	495.6 ± 74.4	16.7 ± 2.3	61.9 ± 8.5	8.0 ± 1.0
**4**	152.1 ± 22.8	535.6 ± 80.3	28.3 ± 3.7	99.6 ± 13.0	5.4 ± 0.7
**7**	155.0 ± 23.2	574.1 ± 85.9	74.5 ± 11.1	275.9 ± 41.1	2.1 ± 0.3
**9**	269.6 ± 35.0	554.7 ± 72.0	2.4 ± 0.3	4.9 ± 0.6	112.3 ± 13.5
**10**	156.4 ± 18.8	297.3 ± 35.7	6.6 ± 0.9	12.5 ± 1.7	23.7 ± 3.1
**11**	250.2 ± 30.0	1025.4 ± 123.0	6.7 ± 0.9	27.4 ± 3.7	37.3 ± 5.2
**13**	199.4 ± 23.9	424.3 ± 50.1	2.6 ± 0.3	5.5 ± 0.6	76.7 ± 9.9
**14**	162.4 ± 17.9	316.0 ± 34.8	8.6 ± 1.2	16.7 ± 2.3	18.9 ± 2.8
Acyclovir	>1000	NA *	NA *	NA *	NA *

* NA—not active.

**Table 3 ijms-25-04142-t003:** Virucidal activity of polyphenolic compounds from *M. amurensis* heartwood (RT-PCR assay) against HSV-1.

Compounds	Concentration		Ct	−ΔCt	2^−ΔCt^	log_10_
	µg/mL	µM				
**1**	10		24.6 ± 2.9 *	−6.5 ± 0.8	0.0110 ± 0.0014	−1.9 ± 0.2 *
	1		20.4 ± 2.8	−2.3 ± 0.2	0.203 ± 0.024	−0.7 ± 0.09
	0.1		18.1 ± 2.2	0	1.0	0
**2**	10		22.4 ± 2.4 *	−4.3 ± 0.5	0.0508 ± 0.0061	−1.3 ± 0.2 *
	1		18.4 ± 2.4	−0.30 ± 0.03	0.812 ± 0.110	−0.09 ± 0.01
	0.1		18.1 ± 2.2	0	1.0	0
**4**	10		20.7 ± 2.3	−2.6 ± 0.3	0.165 ± 0.021	−0.8 ± 0.1
	1		19.0 ± 2.1	−0.9 ± 0.1	0.536 ± 0.064	−0.30 ± 0.04
	0.1		18.1 ± 2.2	0	1.0	0
**7**	10		19.8 ± 2.4	−1.75 ± 0.20	0.297 ± 0.04	−0.50 ± 0.06
	1		18.3 ± 2.2	−0.20 ± 0.02	0.870 ± 0.11	−0.060 ± 0.008
	0.1		18.1 ± 2.2	0	1.0	0
**9**	10		33.3 ± 4.3 *	−15.2 ± 1.8	0.0000266 ± 0.000003	−4.6 ± 0.6 *
	1		25.9 ± 2.8 *	−7.8 ± 0.9	0.00449 ± 0.0006	−2.3 ± 0.3 *
	0.1		18.6 ± 2.4	−0.50 ± 0.05	0.707 ± 0.092	−0.15 ± 0.02
**10**	10		28.0 ± 3.9 *	−9.9 ± 1.1	0.00105 ± 0.0001	−2.9 ± 0.4 *
	1		20.2 ± 2.6	−2.1 ± 0.3	0.233 ± 0.03	−0.6 ± 0.08
	0.1		18.4 ± 2.4	−0.300 ± 0.03	0.812 ± 0.11	−0.09 ± 0.01
**11**	10		30.2 ± 3.9 *	−12.1 ± 1.4	0.000228 ± 0.0003	−3.6 ± 0.5 *
	1		24.5 ± 3.4 *	−6.4 ± 0.8	0.0118 ± 0.0015	−1.9 ± 0.2 *
	0.1		18.4 ± 2.4	−0.30 ± 0.03	0.812 ± 0.11	−0.09 ± 0.01
**13**	10		31.2 ± 3.7 *	−13.1 ± 1.6	0.000114 ± 0.00001	−3.9 ± 0.5 *
	1		25.8 ± 3.3 *	−7.7 ± 0.8	0.00481 ± 0.0006	−2.3 ± 0.3 *
	0.1		18.6 ± 2.4	−0.50 ± 0.05	0.707 ± 0.092	−0.15 ± 0.02
**14**	10		27.1 ± 3.8 *	−9.0 ± 1.2	0.00195 ± 0.00020	−2.7 ± 0.3 *
	1		19.2 ± 2.5	−1.1 ± 0.1	0.466 ± 0.056	−0.33 ± 0.04
	0.1		18.4 ± 2.4	−0.30 ± 0.03	0.812 ± 0.110	−0.09 ± 0.01
Virus control (DMSO)			18.1 ± 2.2	0	1.0	0
Cell control (DMSO)			≥37.0			

* *p* ≤ 0.05 compared to virus control.

**Table 4 ijms-25-04142-t004:** The effect of polyphenols **9** and **13** on the early stages of HSV-1 infection.

Compounds	CC_50_ (µg/mL)	Pretreatment of Cells			Simultaneous Treatment *			Treatment of Infected Cells		
		IC_50_ (µg/mL)	IC_50_ (µM/mL)	SI	IC_50_ (µg/mL)	IC_50_ (µM/mL)	SI	IC_50_ (µg/mL)	IC_50_ (µM/mL)	SI
**9**	269.6 ± 35.0	70.3 ± 9.1	144.6 ± 18.7	3.8 ± 0.5	17.6 ± 2.3	36.2 ± 4.7	15.3 ± 1.9	24.5 ± 3.2	50.4 ± 6.6	11.0 ± 1.4
**13**	199.4 ± 23.9	47.2 ± 6.1	100.4 ± 13.0	4.2 ± 0.6	13.8 ± 1.6	29.4 ± 3.4	14.4 ± 1.7	20.2 ± 2.8	43.0 ± 6.0	9.8 ± 1.2
Acyclovir	>1000	NA			2.1 ± 0.3	9.3 ± 1.3	430	0.4 ± 0.05	1.8 ± 0.2	>2500

* These data were published by us in [22].

## Data Availability

The data are contained within the article and Appendix A (Reference [38] is cited in the Appendix A).

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
