# Peer review of "Polyphenols from Maackia amurensis Heartwood Protect Neuronal Cells from Oxidative Stress and Prevent Herpetic Infection"

_ijms, 2024, doi:10.3390/ijms25084142_

Round 1

Reviewer 1 Report

Comments and Suggestions for Authors

The article is one of the series article written by the authors and dedicated to the drug Maksar available only on the Russian market.

The methodology of the study is very well planned and conducted which ensures a high standard of the study. The studied compounds very isolated from Maackia extract and described by the NMR and theoretical approach. The cell assays were performed on the Neuro-2a and Vero cells as well as HSV-1 virus and the deep analysis of neuroprotective and anti-HSV-1 activity was correctly and comprehensively performed. The statistical analysis of the experimental data was properly done, and the final conlusions are correct. My question concern the compounds no. 11 and and 13 which were previously isolated and disscussed in: https://www.mdpi.com/1420-3049/28/6/2593. It should be mentioned in the Introduction and clearly described what was previously published and what is the novelty in the case of these two compounds in the manuscript ref. [22].

Author Response

Here we determined the absolute configurations of the asymmetric centers in compounds 10 and 13 isolated from Maackia amurensis, which was not done before.

In our previous report [https://www.mdpi.com/1420-3049/28/6/2593] we used only a model of PQ-induced neurotoxicity. Here compounds 10, 11 and 13 were tested for neuroprotective activity using a model of 6-OHDA-induced neurotoxicity.

Here we also studied the virucidal activity of compounds 10, 11 and 13. In order to study the mechanism of action of compound 13, two additional schemes of treatment of the virus and Vero cells with polyphenols were applied: the compound was added before virus infection (pretreatment of cells) and after penetration of the virus into host cells (treatment of infected cells).

In our previous report [https://www.mdpi.com/1420-3049/28/6/2593] we used a different method of exposure, when Vero cells were treated with polyphenolic compounds and the virus simultaneously.

All these facts are described in Results and Discussion.

Reviewer 2 Report

Comments and Suggestions for Authors

In this work, 14 compounds isolated from the leguminous woody plant Maackia amurensis were tested for their antiviral and neuroprotective activity. The authors identified the structure of the isolated polyphenols and defined the absolute configuration of two of them. The neuroprotective activity of all compounds was evaluated in a mouse neuroblast cell line (Neuro-2a). Of note, the antiviral activity of this compounds against herpes simplex virus type 1 (HSV-1) had already been studied by the authors in a previous work. However, in the present work the authors initiated to characterize their mechanism of action using different techniques. Overall, this work is both original and of interest to the scientific community. However, some issues should be addressed before publication. Therefore, I recommend that this manuscript be accepted after a major revision.

General concerns:

1- The overall quality of the figures should be improved in order to make them easier for the reader to understand. Specifically, in Fig. 4, 5, 6, 7 the columns are not centered with their names and some (*) are inside the columns.

2- Paragraph 2.3 should be merged with 2.4, as well as 2.9 with 2.10, 4.5 with 4.11 and 4.6 with 4.12.

3- It would be better to move the conclusions paragraph immediately after the discussion.

Introduction:

Row 70: add reference.

Results:

Results 2.3: The cytotoxic activity of polyphenolic compounds should be included in the results, at least as a supplementary figure.

Results 2.4:

-          Row 163: The authors should specify whether the polyphenolic compounds treatment occurred before (pretreatment), at the same time or after the treatment with 6-OHDA.

-          Compounds 8 and 12 were also able to significantly increase the cell viability at 1 μM and 0,1-1 uM, respectively. These data should be reported in the text.

-          All compounds were tested at three different increasing concentrations, however none of them appeared to have a dose-dependent effect. Authors should comment on this phenomenon in the text.

Results 2.5: As written by the authors, compound 3 at a concentration of 10 μM wasn’t able to significantly reduce ROS formation. However, it showed strong activity at a concentration of 0,1 uM. should be commented

Results 2.7: row 210, a cell viability increase of 17% and 15% can’t be defined as “high”.

Results 2.10: Commonly in this type of experiments the CPE/concentrations graph for each compound is reported. Since the authors should already have all the necessary data it would not be difficult add the aforementioned graphs in the work, which would be highly informative for readers and would enrich the paper.

Results 2.11: In order to quantify the viral genome in the supernatant the authors should have performed a quantitative PCR using standards with a known viral genome copy number. Alternatively, a comparative RT-PCR is also acceptable but it would be useful to include dose-response graphs in addition to Table 3.

Row 286: it is the paragraph 2.12, not 3.12

Results 2.12: see point “Results 2.10”.

Discussion:

Row 314: 6-OHDA and PQ treatment in cell lines induces a general neurotoxicity. Treatment of non-human cell line with these molecules can’t mimic the complexity of Parkinson’s disease.

Row 353: The authors have no data to propose this hypothesis since the antiviral activity of the compounds has not been tested on any other virus. Virucidal activity could be due to many factors, including destruction of the viral envelope. The discussion on possible mechanisms of action should be conducted more thoroughly in the paper.

Row 355: Why did the authors not measure HSV-1 induced ROS production in the presence of polyphenolic compounds? It would be a very interesting aspect to explore further.

Materials and Methods:

MeM 4.7-4.8: In the cell viability test (4.7) a concentration of 100 μM of 6-OHDA was used while a concentration of 120 μM was used for the ROS analysis experiments (4.8). Why were different concentrations used in the two protocols?

MeM 4.13: According to the described protocol the compounds were still present when pretreated HSV-1 was used to infect VERO cells. To specifically evaluate the virucidal activity of the compounds the authors should specify how much the pre-treated virus (and therefore the compounds contained in it) was diluted at the time of infection. If the final concentration of the compound on the cells at the time of infection is greater than 0.1 ug/ml Acv (row 475) should be the absorbance of virus-infected cells treated at the time of infection with the same concentration of compound. Otherwise we cannot speak of virucidal activity.

Comments on the Quality of English Language

Minor editing of English language required

Author Response

Reviewer’s comments

Our response

General concerns:

1- The overall quality of the figures should be improved in order to make them easier for the reader to understand. Specifically, in Fig. 4, 5, 6, 7 the columns are not centered with their names and some (*) are inside the columns.

We have improved the quality of the Figures according to the reviewer’s comments.

2- Paragraph 2.3 should be merged with 2.4, as well as 2.9 with 2.10, 4.5 with 4.11 and 4.6 with 4.12.

We believe it would be more convenient for the readers if we provide this information separately in Materials and Methods and Results sections.

3- It would be better to move the conclusions paragraph immediately after the discussion.

We used the template for IJMS. The structure of the manuscript is in accordance with the template.

Introduction:

Row 70: add reference.

We have made the necessary corrections in the text.

Results:

Results 2.3: The cytotoxic activity of polyphenolic compounds should be included in the results, at least as a supplementary figure.

The cytotoxic activity of polyphenolic compounds against Vero cells is presented in Table 2.

The polyphenolic compounds did not show significant cytotoxic activity against Neuro-2a cells at concentrations £ 100 µM, so we were not able to determine CC50 values.

We did not test the polyphenolic compounds for cytotoxicity at concentrations higher than 100 µM.

Results 2.4:

-          Row 163: The authors should specify whether the polyphenolic compounds treatment occurred before (pretreatment), at the same time or after the treatment with 6-OHDA.

We provided this information in Materials and Methods and added to the Results section as well.

Neuro-2a cells were incubated with the studied polyphenolic compounds for 1h. Then, 100 µM of 6-OHDA were added.

Compounds 8 and 12 were also able to significantly increase the cell viability at 1 μM and 0,1-1 uM, respectively. These data should be reported in the text.

We have reported these data in the text.

-          All compounds were tested at three different increasing concentrations, however none of them appeared to have a dose-dependent effect. Authors should comment on this phenomenon in the text.

The mechanism of action of some polyphenolic compounds is complex and multitarget. One of the reasons for the lack of dose-dependency may be that some of the compounds at high concentration can possibly  inhibit or activate some of the enzymes or signaling pathways in a cell, This can cause a “bell-shaped” or reverse dependence of the dose-response for some polyphenolic compounds.

Results 2.5: As written by the authors, compound 3 at a concentration of 10 μM wasn’t able to significantly reduce ROS formation. However, it showed strong activity at a concentration of 0,1 uM. should be commented

Compound 3 also can possibly inhibit or activate some of the antioxidant enzymes in a cell. Besides, some polyphenols can possess “prooxidant” activity at high concentrations. This can possibly cause a “bell-shaped” dependence of the dose-response.

Results 2.7: row 210, a cell viability increase of 17% and 15% can’t be defined as “high”.

We have made the necessary corrections in the text.

Results 2.10: Commonly in this type of experiments the CPE/concentrations graph for each compound is reported. Since the authors should already have all the necessary data it would not be difficult add the aforementioned graphs in the work, which would be highly informative for readers and would enrich the paper.

The CPE/concentrations graph can be found in Supplementary Material (Figure S8).

Results 2.11: In order to quantify the viral genome in the supernatant the authors should have performed a quantitative PCR using standards with a known viral genome copy number. Alternatively, a comparative RT-PCR is also acceptable but it would be useful to include dose-response graphs in addition to Table 3.

In this study we performed RT-PCR analysis. The dose-response graphs can be found in Supplementary Material (Figure S9).

-     Row 286: it is the paragraph 2.12, not 3.12

We have made the necessary corrections in the text.

Results 2.12: see point “Results 2.10”.

We consider the IC50 value as the main characteristic of the virucidal activity of polyphenolic compounds. We calculated 50% inhibitory concentration (the concentration of the compound that reduced the virus-induced CPE by 50%, IC50) using a regression analysis of the dose–response curve (CPE/concentrations graphs).

We consider that Tables 2 and 4 and CPE/concentrations graphs contain the same information. However, we provided the graphs for virucidal activity just as an illustration in Supplementary Material (Figure S8).

Discussion:

Row 314: 6-OHDA and PQ treatment in cell lines induces a general neurotoxicity. Treatment of non-human cell line with these molecules can’t mimic the complexity of Parkinson’s disease.

We have made the necessary corrections in the text.

Notably, the model we used is generally accepted model of Parkinson’s disease [Bove, J.; Prou, D.; Perier, C.; Przedborski, S. Toxin-induced models of Parkinson’s disease. NeuroRX 2005, 2, 484–494.].

Row 353: The authors have no data to propose this hypothesis since the antiviral activity of the compounds has not been tested on any other virus. Virucidal activity could be due to many factors, including destruction of the viral envelope. The discussion on possible mechanisms of action should be conducted more thoroughly in the paper.

Here the virus was pretreated with the polyphenolic compounds before infection. This method of exposure was designed to find out whether the compounds can interact with the viral envelope [Figueiredo, G.G.; Coronel, O.A.; Trabuco, A.C.; Bazán, D.E.; Russo, R.R.; Alvarenga, N.L.; Aquino V.H. Steroidal saponins from the roots of Solanum sisymbriifolium Lam. (Solanaceae) have inhibitory activity against dengue virus and yellow fever virus. Braz J Med Biol Res. 2021; 54(7): e10240. doi: 10.1590/1414-431X2020e10240; Tai C.J., Li C.L., Tai C.J., Wang C.K., Lin L.T. Early Viral Entry Assays for the Identification and Evaluation of Antiviral Compounds. J. Vis. Exp. 2015;104:e53124. doi: 10.3791/53124]. The main components of the viral envelope of HSV-1 are viral proteins (gD, gB, gH/gL) [Akhtar J., Shukla D. Viral entry mechanisms: Cellular and viral mediators of herpes simplex virus entry. FEBS J. 2009;276:7228–7236. doi: 10.1111/j.1742-4658.2009.07402.x.; Arii J., Kawaguchi Y. The Role of HSV Glycoproteins in Mediating Cell Entry. In: Kawaguchi Y., Mori Y., Kimura H., editors. Human Herpesviruses. Advances in Experimental Medicine and Biology. Volume 1045. Springer; Singapore: 2018. pp. 3–21.].

Row 355: Why did the authors not measure HSV-1 induced ROS production in the presence of polyphenolic compounds? It would be a very interesting aspect to explore further.

This is going to be the next stage of our investigation.

Materials and Methods:

MeM 4.7-4.8: In the cell viability test (4.7) a concentration of 100 μM of 6-OHDA was used while a concentration of 120 μM was used for the ROS analysis experiments (4.8). Why were different concentrations used in the two protocols?

The concentrations were chosen experimentally.

100 μM and 120 μM of 6-OHDA caused optimal decrease in cell viability and increase of ROS level, respectively.

MeM 4.13: According to the described protocol the compounds were still present when pretreated HSV-1 was used to infect VERO cells. To specifically evaluate the virucidal activity of the compounds the authors should specify how much the pre-treated virus (and therefore the compounds contained in it) was diluted at the time of infection. If the final concentration of the compound on the cells at the time of infection is greater than 0.1 ug/ml Acv (row 475) should be the absorbance of virus-infected cells treated at the time of infection with the same concentration of compound. Otherwise we cannot speak of virucidal activity.

To study the effect of polyphenols on the different stages of the HSV-1 life cycle we used one of the application schemes - pretreatment of the virus with compounds. This allowed us to determine the direct effect of polyphenols on viral particles, according to the method for determining the virucidal effect described by a number of authors:

1) Figueiredo, G.G.; Coronel, O.A.; Trabuco, A.C.; Bazán, D.E.; Russo, R.R.; Alvarenga, N.L.; Aquino V.H. Steroidal saponins from the roots of Solanum sisymbriifolium Lam. (Solanaceae) have inhibitory activity against dengue virus and yellow fever virus. Braz J Med Biol Res. 2021; 54(7): e10240. doi: 10.1590/1414-431X2020e10240

2) Low ZX, OuYong BM, Hassandarvish P, Poh CL, Ramanathan B. Antiviral activity of silymarin and baicalein against dengue virus. Sci Rep. 2021 Oct 27;11(1):21221. doi: 10.1038/s41598-021-98949-y. PMID: 34707245; PMCID: PMC8551334.

3) Fernandes LS, da Silva ML, Dias RS, da S Lucindo MS, da Silva ÍEP, Silva CC, Teixeira RR, de Paula SO. Evaluation of Antiviral Activity of Cyclic Ketones against Mayaro Virus. Viruses. 2021 Oct 21;13(11):2123. doi: 10.3390/v13112123. PMID: 34834929; PMCID: PMC8625987

Round 2

Reviewer 2 Report

Comments and Suggestions for Authors

The manuscript has been improved with revision. However, minor revisions are still required before publication.

1) The Authors wrote this two sentences in response to the first revision.

The mechanism of action of some polyphenolic compounds is complex and multitarget. One of the reasons for the lack of dose-dependency may be that some of the compounds at high concentration can possibly inhibit or activate some of the enzymes or signaling pathways in a cell, This can cause a “bell-shaped” or reverse dependence of the dose-response for some polyphenolic compounds

Compound 3 also can possibly inhibit or activate some of the antioxidant enzymes in a cell. Besides, some polyphenols can possess “prooxidant” activity at high concentrations. This can possibly cause a “bell-shaped” dependence of the dose-response”.

Both sentences are really informative, therefore I suggest to include them in the main text of the paper.  

2) Figure S8 and Figure S9 should be mentioned in the results.

3) Paragraphs 4.6 and 4.12 describe the same protocol “Cytotoxicity of the tested compound”. Therefore, I suggest to merge the paragraphs.

Paragraphs 4.6 and 4.12 describe the same protocol “Cell culture condition”. Therefore, I suggest to merge the paragraphs.

Paragraph 2.3 is just one sentence. In order to make the work more readable and avoid unnecessary division into paragraphs, I suggest combining 2.3 with the following paragraph.

Paragraph 2.9 is just one sentence. In order to make the work more readable and avoid unnecessary division into paragraphs, I suggest combining 2.9 with the following paragraph.

4) As answer to the first revision the Authors wrote: “Here the virus was pretreated with the polyphenolic compounds before infection. This method of exposure was designed to find out whether the compounds can interact with the viral envelope.”

I agree with this statement, the virucidal activity is mediated by an interaction of the compounds with the viral envelope. However, the viral envelope is a lipid bilayer studded with an outer layer of virus envelope glycoproteins. Compounds interaction with viral envelop could have many different outcome: 1) interaction/disruption of lipid bilayer; 2) unspecific denaturation/inactivation of viral glycoprotein; 3) specific interaction with glycoprotein binding site.

Therefore, authors have no data to say that the virucidal activity may be due only to the ability of polyphenols to specifically bind the surface glycoproteins of HSV-1 competing for binding sites of the proteins with cellular receptors (row 351-353). If the authors want to address this issue under discussion they should also include the other possibilities.

Author Response

Reviewer’s comments

Our response

1) The Authors wrote this two sentences in response to the first revision.

The mechanism of action of some polyphenolic compounds is complex and multitarget. One of the reasons for the lack of dose-dependency may be that some of the compounds at high concentration can possibly inhibit or activate some of the enzymes or signaling pathways in a cell, This can cause a “bell-shaped” or reverse dependence of the dose-response for some polyphenolic compounds

Compound 3 also can possibly inhibit or activate some of the antioxidant enzymes in a cell. Besides, some polyphenols can possess “prooxidant” activity at high concentrations. This can possibly cause a “bell-shaped” dependence of the dose-response”.

Both sentences are really informative, therefore I suggest to include them in the main text of the paper. 

We added these sentences to the Discussion section.

2) Figure S8 and Figure S9 should be mentioned in the results.

We added these references to the text in the Results Section.

 3) Paragraphs 4.6 and 4.12 describe the same protocol “Cytotoxicity of the tested compound”. Therefore, I suggest to merge the paragraphs.

Paragraphs 4.6 and 4.12 describe the same protocol “Cell culture condition”. Therefore, I suggest to merge the paragraphs.

Paragraph 2.3 is just one sentence. In order to make the work more readable and avoid unnecessary division into paragraphs, I suggest combining 2.3 with the following paragraph.

Paragraph 2.9 is just one sentence. In order to make the work more readable and avoid unnecessary division into paragraphs, I suggest combining 2.9 with the following paragraph.

We have merged these paragraphs 2.3 and 2.4, 2.9 and 2.10. We believe it would better to merge paragraphs 4.5 and 4.6, 4.11 and 4.12, so that we do not mix different type of cells.

4) As answer to the first revision the Authors wrote: “Here the virus was pretreated with the polyphenolic compounds before infection. This method of exposure was designed to find out whether the compounds can interact with the viral envelope.”

I agree with this statement, the virucidal activity is mediated by an interaction of the compounds with the viral envelope. However, the viral envelope is a lipid bilayer studded with an outer layer of virus envelope glycoproteins. Compounds interaction with viral envelop could have many different outcome: 1) interaction/disruption of lipid bilayer; 2) unspecific denaturation/inactivation of viral glycoprotein; 3) specific interaction with glycoprotein binding site.

Therefore, authors have no data to say that the virucidal activity may be due only to the ability of polyphenols to specifically bind the surface glycoproteins of HSV-1 competing for binding sites of the proteins with cellular receptors (row 351-353). If the authors want to address this issue under discussion they should also include the other possibilities.

In order to be more thorough in our Discussion we have modified this sentence and wrote:

“This activity may be due to the ability of these polyphenols to interact with the surface glycoproteins of HSV-1 or lipid components of the viral envelope.”
